# Trends and determinants of length of stay and hospital reimbursement following knee and hip replacement: evidence from linked primary care and NHS hospital records from 1997 to 2014

Edward Burn,[1] Christopher J Edwards,[2] David W Murray,[1] Alan Silman,[1] Cyrus Cooper,[1,3] Nigel K Arden,[1,3] Rafael Pinedo-Villanueva,[1,3] Daniel Prieto-Alhambra[1,4]

[1]Nuffield Department of Orthopaedics, Rheumatology and Musculoskeletal Sciences, University of Oxford, Oxford, UK
[2]NIHR Wellcome Trust Clinical Research Facility, University Hospital Southampton, Southampton, UK
[3]MRC Lifecourse Epidemiology Unit, Southampton University, Southampton, UK
[4]GREMPAL Research Group, Idiap Jordi Gol and CIBERFes, Universitat Autonoma de Barcelona and Instituto de Salud Carlos III, Barcelona, Spain

**Correspondence to**
Dr Rafael Pinedo-Villanueva;
rafael.pinedo@ndorms.ox.ac.uk

## ABSTRACT

**Objectives** To measure changes in length of stay following total knee and hip replacement (TKR and THR) between 1997 and 2014 and estimate the impact on hospital reimbursement, all else being equal. Further, to assess the degree to which observed trends can be explained by improved efficiency or changes in patient profiles.

**Design** Cross-sectional study using routinely collected data.

**Setting** National Health Service primary care records from 1995 to 2014 in the Clinical Practice Research Datalink were linked to hospital inpatient data from 1997 to 2014 in Hospital Episode Statistics Admitted Patient Care.

**Participants** Study participants had a diagnosis of osteoarthritis or rheumatoid arthritis.

**Interventions** Primary TKR, primary THR, revision TKR and revision THR.

**Primary outcome measures** Length of stay and hospital reimbursement.

**Results** 10 260 primary TKR, 10 961 primary THR, 505 revision TKR and 633 revision THR were included. Expected length of stay fell from 16.0 days (95% CI 14.9 to 17.2) in 1997 to 5.4 (5.2 to 5.6) in 2014 for primary TKR and from 14.4 (13.7 to 15.0) to 5.6 (5.4 to 5.8) for primary THR, leading to savings of £1537 and £1412, respectively. Length of stay fell from 29.8 (17.5 to 50.5) to 11.0 (8.3 to 14.6) for revision TKR and from 18.3 (11.6 to 28.9) to 12.5 (9.3 to 16.8) for revision THR, but no significant reduction in reimbursement was estimated. The estimated effect of year of surgery remained similar when patient characteristics were included.

**Conclusions** Length of stay for joint replacement fell substantially from 1997 to 2014. These reductions have translated into substantial savings. While patient characteristics affect length of stay and reimbursement, patient profiles have remained broadly stable over time. The observed reductions appear to be mostly explained by improved efficiency.

### Strengths and limitations of this study

► Routinely collected data provided real-world information on trends in length of stay following primary knee and hip replacement and revision procedures.
► Patient characteristics were controlled for to assess whether trends in length of stay and associated hospital reimbursement were explained by changes in patient characteristics or improved efficiency.
► Codes used to identify diagnoses of osteoarthritis have not been fully evaluated.

end-stage arthritis, leading to substantial gains in an individual's quality of life.[1 2] The procedures are a cost-effective use of healthcare resources, with their costs justified by the expected health gains for the individuals receiving them.[1 2] Rates of joint replacement grew substantially between 1991 and 2006 in the UK.[3] Demand is projected to continue to rise in the USA up to 2030 and, in turn, the number of revision procedures is also expected to increase.[4]

Satisfying rising demand will require either additional resources, which is unlikely given current funding constraints, or reducing the cost of the procedures. Length of stay is a key component of the overall cost of joint replacement. Greater efficiency should reduce the expected length of stay for an individual requiring surgery and lead to cost-savings, all else being equal. If patient health outcomes are not compromised in the process, greater efficiency would make joint replacement a more cost-effective procedure.

Length of stay following joint replacement has fallen substantially.[5–11] Although such reductions may be explained by improved

## INTRODUCTION

Knee and hip replacements relieve pain and improve function for individuals with

efficiency, they may also be explained by changes in patient profiles over time. As patient characteristics are likely to influence length of stay, changes in these characteristics would partially explain the observed changes in stay length. Individuals at greater risk of poor health outcomes, due to either preoperative comorbidities or postoperative complications, can be expected to have longer stays. Social determinants can also be expected to influence length of stay.

We assessed the extent to which length of stay following joint replacement decreased in the English National Health Service (NHS) from 1997 to 2014, and whether such changes have been translated into reduced reimbursement to hospitals, all else being equal. We considered the degree to which observed changes can be explained by improved efficiency or changes in patient profiles.

## METHODS
### Study design
Individuals with a diagnosis of osteoarthritis (OA) or rheumatoid arthritis (RA) were identified from primary care records. Linked hospital records were used to identify the procedures of interest following diagnosis: primary total knee replacement (TKR), primary total hip replacement (THR), revision TKR and revision THR. Length of stay and hospital reimbursement for these surgeries were calculated. Trends in length of stay over time were assessed using univariable generalised linear models. How such reductions have translated into lower hospital reimbursement, all else being equal, was assessed by examining trends in reimbursement, at current rates, in a similar manner. Patient characteristics were added as explanatory variables to the models to consider whether trends were explained by changes in efficiency or changes in patient profiles.

### Setting
Primary care records were derived from the Clinical Practice Research Datalink (CPRD). CPRD contains medical records and demographic details for around 7% of the UK population, with those included representative of the general population.[12] A subset of practices within CPRD were linked to inpatient hospital records, provided by Hospital Episode Statistics Admitted Patient Care (HES APC). HES APC contains data on all NHS hospital admissions in England, with each uninterrupted inpatient stay at one hospital recorded as a spell (the time a patient spent in a particular hospital).[13] The CPRD extract used to identify diagnoses began in January 1995. The HES APC extract used to identify events of interest began in April 1997. Linked data from both databases were extracted up to March 2014.

### Participants
We extracted CPRD records for individuals with a diagnosis of either OA or RA. Study participants were identified

separately for procedures relating to the knee and hip. For procedures relating to the knee, CPRD records were extracted for those individuals with a diagnosis recorded for RA or knee OA. For procedures relating to the hip, records were extracted for individuals with a diagnosis of RA or hip OA.

Some individuals had diagnostic codes for both RA and OA. It was not possible to determine which of their diagnoses drove the decision for joint replacement. The need for a joint replacement for those with RA is often dependent on the development of secondary OA. Diagnosis of RA was therefore taken as the index diagnosis from which follow-up started when patients had both diagnoses recorded.

CPRD records were then linked with HES APC data and procedures of interest after the index diagnosis were identified. Only the first instance of each procedure for an individual was included in the analysis, and any bilateral procedures were excluded. Although no individuals contributed multiple records of the same surgery, an individual could provide data on multiple procedures.

### Variables
Diagnoses of RA, knee OA and hip OA were identified using clinical read codes within CPRD. Events of interest (TKR, TKR revision, THR and THR revision) were identified using operating procedure codes (OPCS) in HES APC.

The date of a procedure recorded in HES APC was used to identify the year of surgery and, given an individual's year of birth recorded in CPRD, age at surgery. Gender was recorded in CPRD. Diagnosis codes in HES APC were used to identify diseases included in the Royal College of Surgeons (RCS) Charlson score, a measure of comorbidities, and to calculate the overall summary score (0, 1, 2 or 3+).[14] Index of Multiple Deprivation (IMD) quintiles, a measure of socioeconomic status (with IMD quintile 5 implying the highest level of deprivation), were derived from CPRD.

Length of stay was recorded within HES APC by days for the spell in which a procedure occurred. Hospital spells were assigned to Healthcare Resource Groups (HRGs), which group clinically similar treatments using common levels of healthcare resources. The 2017/2018 draft prices were used to estimate hospital reimbursement of spells based on the HRG, the method of admission (elective or non-elective), the expected physiotherapy cost and any stay longer than the trim-point, which is the upper limit of length of stay incorporated in the HRG cost. While variation between individuals' hospital reimbursement were influenced by HRG codes and type of admission, high estimates were driven primarily by stays longer than the trim-point. Changes in reimbursement over time, therefore, reflect the effect of changing length of stay on reimbursement to hospitals from the NHS at current levels, not the accounting costs incurred by hospitals or historical reimbursement rates. Secondary diagnosis codes that were ineligible for grouping were dropped

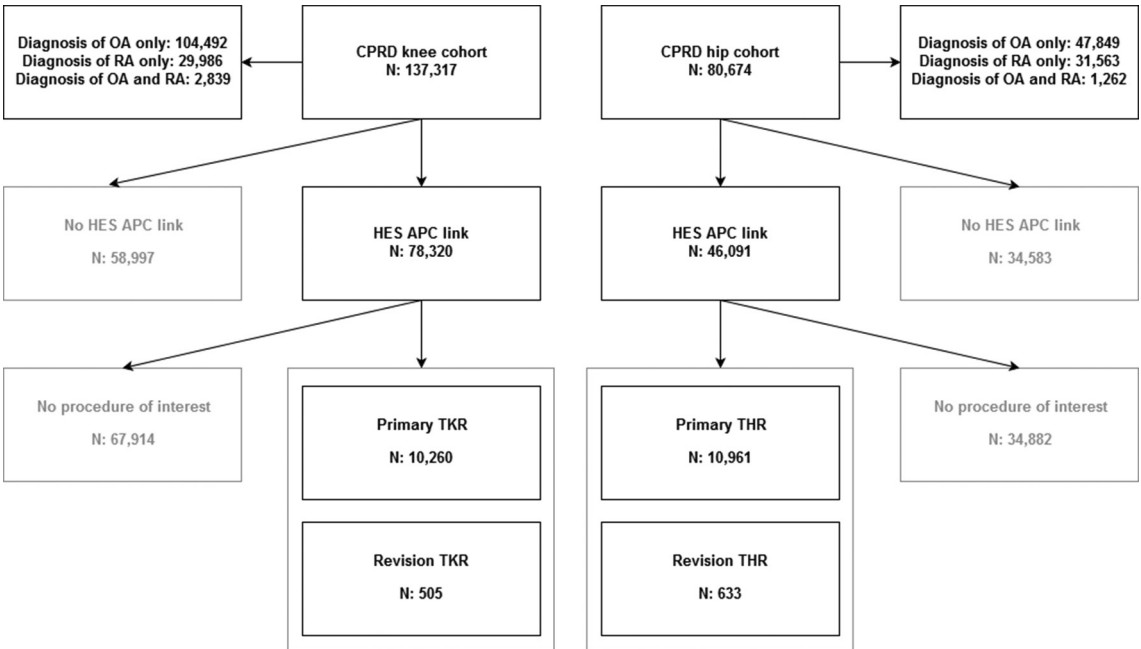

**Figure 1** Study inclusion flow chart. CPRD, Clinical Practice Research Datalink; HES APC, Hospital Episode Statistics Admitted Patient Care; OA, osteoarthritis; RA, rheumatoid arthritis; THR, total hip replacement; TKR, total knee replacement.

from the record. Reimbursement was classified as missing if the primary diagnosis or any other variable required for the grouper software was ineligible.

## Statistical methods

Trends in length of stay and hospital reimbursement over time for each procedure were assessed using univariable regression models that used the year of surgery as the explanatory factor. Multivariable regressions that included patient characteristics were also specified. We compared the estimated effect of year of surgery on length of stay and hospital reimbursement with and without patient characteristics to determine the extent to which a trend over time could be explained by change in patient characteristics or improvements in efficiency. A similar approach has previously been used to assess whether variations in length of stay between types of providers are due to differences in efficiency or patient selection.[15]

As both length of stay and hospital reimbursement are non-negative and right-skewed, generalised linear models with a gamma distribution and a log-link were specified.[16] For continuous variables, non-linearity was incorporated using restricted cubic splines. Interactions between year and patient characteristics were tested for in the multivariable model through a comparison, using analysis of variance, of models with and without interaction terms. The Bayesian information criterion (BIC) was used to decide whether to specify a variable's relationship as non-linear and, if included, how many knots to include. Compared with the Akaike information criterion, BIC favours parsimony and has been recommended for explanatory models.[17]

Year of surgery, diagnosis (RA or OA), age, gender, RCS Charlson score and IMD quintile were included as explanatory variables in each of the multivariable models. The code for RA was omitted from calculations of the RCS Charlson score as RA was specified as a separate variable. The estimated effect of the RCS Charlson score is that of comorbidities aside from RA. The two highest RCS Charlson score categories, 2 and 3+, were combined into the category 2+ as there were few scores of 3 or more.

The only explanatory variable with missing data was the IMD quintile, which was missing for nine TKR and seven THR; these individuals were omitted from all regressions. There were no missing data for length of stay. Reimbursement estimates could not be calculated for 2.3% of the spells for which the HRG grouper did not produce a valid HRG code. Spells without HRG codes were dropped from the regressions on hospital reimbursement, but were included in the analysis of length of stay.

Data analysis was primarily undertaken in R V.3.3.1,[18] using dplyr[19] for data manipulation, rms[20] for fitting regression models and ggplot2[21] to produce plots. HRG4+ codes were derived using the 2015/2016 Reference Costs Grouper.

## RESULTS
### Study participants

Records from 21 128 patients with OA or RA were included in the analysis, with 10 260 undergoing primary TKR, 10 961 primary THR, 505 TKR revision and 633 THR revision in the study period. Inclusion of study participants is described in a flow chart in figure 1. Around 2% of individuals had diagnoses of both OA and RA. RA was taken as the index diagnosis for these individuals.

**Table 1** Patient characteristics by procedure

| | Primary TKR | Primary TKR | Revision TKR | Revision THR |
|---|---|---|---|---|
| No of study participants | 10260 | 10961 | 505 | 633 |
| Age (mean (SD)) | 70.01 (9.24) | 68.93 (10.53) | 69.43 (9.93) | 70.34 (10.91) |
| Gender: male (n (%)) | 4426 (43.1) | 4525 (41.3) | 233 (46.1) | 264 (41.7) |
| Year of surgery (median (IQR)) | 2009 (2006, 2011) | 2008 (2005, 2011) | 2009 (2007, 2011) | 2009 (2006, 2011) |
| Diagnoses recorded in CPRD | | | | |
| Rheumatoid arthritis (n (%)) | 851 (8.3) | 639 (5.8) | 65 (12.9) | 92 (14.5) |
| Diagnoses recorded in HES APC | | | | |
| Myocardial infarction (n (%)) | 138 (1.3) | 140 (1.3) | 6 (1.2) | 14 (2.2) |
| Congestive cardiac failure (n (%)) | 96 (0.9) | 137 (1.2) | 8 (1.6) | 14 (2.2) |
| Peripheral vascular disease (n (%)) | 101 (1.0) | 102 (0.9) | 10 (2.0) | 8 (1.3) |
| Cerebrovascular disease (n (%)) | 54 (0.5) | 56 (0.5) | 3 (0.6) | 5 (0.8) |
| Dementia (n (%)) | 25 (0.2) | 37 (0.3) | 0 (0.0) | 10 (1.6) |
| Chronic pulmonary disease (n (%)) | 1091 (10.6) | 998 (9.1) | 58 (11.5) | 55 (8.7) |
| Rheumatological disease (n (%)) | 593 (5.8) | 474 (4.3) | 38 (7.5) | 50 (7.9) |
| Liver disease (n (%)) | 20 (0.2) | 22 (0.2) | 0 (0.0) | 5 (0.8) |
| Diabetes mellitus (n (%)) | 991 (9.7) | 755 (6.9) | 56 (11.1) | 45 (7.1) |
| Haemiplegia or paraplegia (n (%)) | 18 (0.2) | 17 (0.2) | 4 (0.8) | 5 (0.8) |
| Renal disease (n (%)) | 202 (2.0) | 235 (2.1) | 11 (2.2) | 20 (3.2) |
| Any malignancy (n (%)) | 78 (0.8) | 97 (0.9) | 7 (1.4) | 9 (1.4) |
| Metastatic solid tumour (n (%)) | 6 (0.1) | 10 (0.1) | 1 (0.2) | 2 (0.3) |
| AIDS HIV infection (n (%)) | 0 (0.0) | 1 (0.0) | 0 (0.0) | 0 (0.0) |
| RCS Charlson score (n (%)) | | | | |
| 0 | 7443 (72.5) | 8413 (76.8) | 342 (67.7) | 455 (71.9) |
| 1 | 2302 (22.4) | 2110 (19.3) | 130 (25.7) | 124 (19.6) |
| 2 | 437 (4.3) | 353 (3.2) | 29 (5.7) | 45 (7.1) |
| 3+ | 78 (0.8) | 85 (0.8) | 4 (0.8) | 9 (1.4) |
| IMD quintiles (n (%)) | | | | |
| 1 | 2330 (22.7) | 2779 (25.4) | 128 (25.3) | 208 (32.9) |
| 2 | 2487 (24.2) | 2794 (25.5) | 120 (23.8) | 155 (24.5) |
| 3 | 2383 (23.2) | 2479 (22.6) | 117 (23.2) | 127 (20.1) |
| 4 | 1930 (18.8) | 1923 (17.5) | 105 (20.8) | 84 (13.3) |
| 5 | 1121 (10.9) | 980 (8.9) | 35 (6.9) | 59 (9.3) |
| Missing | 9 (0.1) | 6 (0.1) | 0 (0.0) | 0 (0.0) |
| Ethnicity (n (%)) | | | | |
| Non-White | 240 (2.3) | 60 (0.5) | 15 (3.0) | 6 (0.9) |
| White | 9073 (88.4) | 9643 (88.0) | 460 (91.1) | 575 (90.8) |
| Missing | 947 (9.2) | 1258 (11.5) | 30 (5.9) | 52 (8.2) |

CPRD, Clinical Practice Research Datalink; HES APC, Hospital Episode Statistics Admitted Patient Care; IMD, Index of Multiple Deprivation; RCS, Royal College of Surgeons; THR, total hip replacement; TKR, total knee replacement.

Individuals' characteristics at the time of surgery are described in table 1 and are stratified by year of surgery in the online supplementary appendix 1. Age, gender and socioeconomic status, measured by IMD quintile, remained relatively stable over the study period. While the proportion of patients with a diagnosis of RA fell over time, the number of comorbidities at surgery, as measured by the RCS Charlson score, increased.

### Trends in expected length of stay and hospital reimbursement over time

Observed length of stay and hospital reimbursement by year of surgery are detailed in the online supplementary

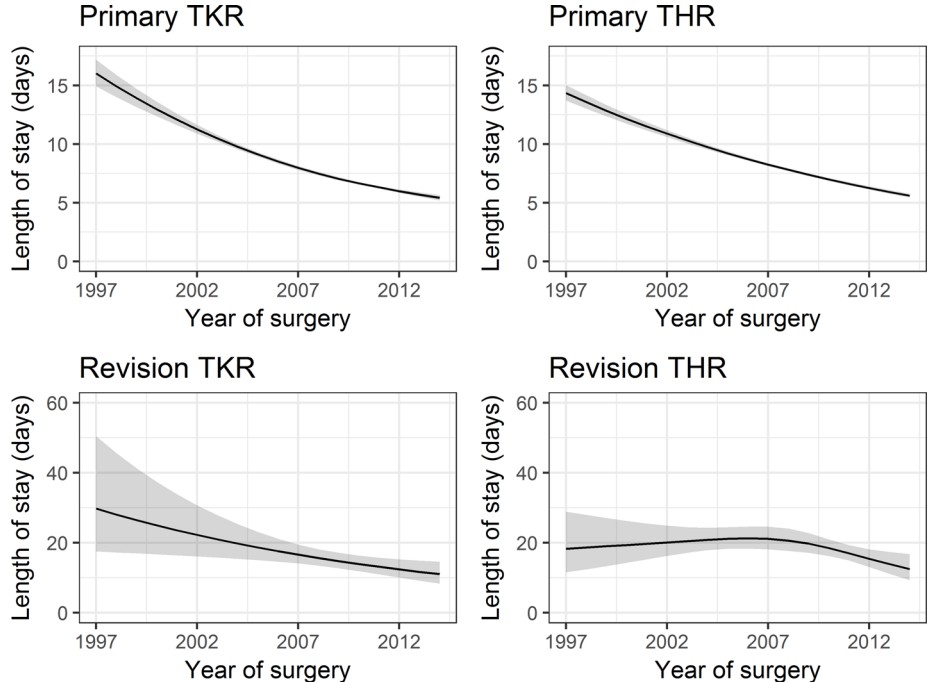

**Figure 2** Trends in length of stay. Estimated effect of year of surgery on length of stay (in days). THR, total hip replacement; TKR, total knee replacement.

appendix 1. The estimated effects of year of surgery on length of stay from univariable regressions are presented in figure 2. Expected length of stay fell from 16.0 days (95% CI 14.9 to 17.2) in 1997 to 5.4 (5.2 to 5.6) in 2014 for primary TKR, from 14.4 (13.7 to 15.0) to 5.6 (5.4 to 5.8) for primary THR, from 29.8 (17.5 to 50.5) to 11.0 (8.3 to 14.6) for revision TKR and from 18.3 (11.6 to 28.9) to 12.5 (9.3 to 16.8) for revision THR. Including patient characteristics in the regression models had little effect on the association between year of surgery and length of stay and hospital reimbursement, as shown in table 2. For example, multivariable adjustment for patient characteristics changed the expected reduction in length of stay for primary THR from 5% (relative effect (RE) 0.95 (95% CI 0.94 to 0.95)) to 6% per year (RE 0.94 (95% CI 0.94 to 0.95)).

Estimated trends in hospital reimbursement from univariable regressions were broadly similar to those estimated for length of stay (figure 3). Expected mean hospital reimbursement fell from £7634 (£7464 to £7808) for hospital stays in 1997 to £6097 (£6016 to £6179) for 2014 stays for primary TKR, from £7300 (£7187 to £7414) to £5888 (£5827 to £5949) for primary THR, from £9088 (£7727 to £10 689) to £7296 (£6717 to £7926) for revision TKR and from £7655 (£6385 to £9177) to £7564 (£6743 to £8485) for revision THR. Including patient characteristics in the regression models did not remove the estimated association between year of surgery and hospital reimbursement, as shown in table 3. For example, the expected reduction in hospital reimbursement for revision TKR was 1% per year (RE: 0. 99 (95% CI 0.97 to 1.00)) in the

unadjusted model and 1% per year (RE: 0. 99 (95% CI 0.98 to 1.00)) in the multivariable model.

## Impact of patient characteristics on expected length of stay and hospital reimbursement

Tables 2 and 3 detail the relative effects of patient characteristics on length of stay and hospital reimbursement from the fully specified models. No interactions between year of surgery and patient characteristics were identified. Higher age at surgery was associated with longer stays for each procedure. Where the effect of age was non-linear, its impact was more pronounced among older patients. Holding all other explanatory variables at their average, expected length of stay increased from 5.7 (5.5 to 5.9) for a 55-year-old to 9.1 (8.7 to 9.5) for an 85-year-old undergoing primary TKR, from 6.2 (5.9 to 6.4) to 10.6 (10.1 to 11.0) for primary THR, from 9.6 (6.5 to 14.1) to 14.4 (9.5 to 21.8) for revision TKR and from 12.4 (9.4 to 16.3) to 27.3 (21.0 to 35.5) for revision THR. Similar upward trends were estimated for hospital reimbursement for each of the procedures of interest. With other explanatory variables held at their average value, expected reimbursement increased from £5856 (£5791 to £5923) for a 55-year-old to £6553 (£6471 to £6636) for an 85-year-old undergoing primary TKR, from £5792 (£5734 to £5850) to £6596 (£6520 to £6672) for primary THR, from £6409 (£5811 to £7069) to £7691 (£6976 to £8480) for revision TKR and from £8071 (£7292 to £8934) to £10 269 (£9315 to £11 320) for revision THR. Plots of the partial effect of age on length of stay and hospital reimbursement are provided in the online supplementary appendix 1.

**Table 2** Multivariable regressions for length of stay

| | Primary TKR (RE (95% CI)) | | Primary THR (RE (95% CI)) | | Revision TKR (RE (95% CI)) | | Revision THR (RE (95% CI)) | |
|---|---|---|---|---|---|---|---|---|
| | Unadjusted | Adjusted | Unadjusted | Adjusted | Unadjusted | Adjusted | Unadjusted | Adjusted |
| Year of surgery | 0.93 (0.92 to 0.94) | 0.93 (0.93 to 0.94) | 0.95 (0.94 to 0.95) | 0.94 (0.94 to 0.95) | 0.94 (0.90 to 0.99) | 0.96 (0.92 to 1.00) | 1.02 (0.96 to 1.07) | 1.02 (0.96 to 1.07) |
| Year of surgery* | 1.01 (1.00 to 1.02) | 1.01 (1.00 to 1.02) | | | | | 0.93 (0.87 to 1.00) | 0.91 (0.85 to 0.98) |
| Age | | 1.00 (1.00 to 1.00) | | 1.00 (1.00 to 1.01) | | 0.99 (0.96 to 1.02) | | 1.03 (1.02 to 1.04) |
| Age* | | 1.02 (1.01 to 1.02) | | 1.02 (1.01 to 1.02) | | 1.03 (0.99 to 1.07) | | |
| Gender: male | | 0.98 (0.95 to 1.00) | | 0.94 (0.91 to 0.96) | | 1.03 (0.78 to 1.37) | | 0.89 (0.72 to 1.09) |
| Diagnosis: RA | | 1.17 (1.11 to 1.22) | | 1.17 (1.10 to 1.25) | | 2.35 (1.56 to 3.68) | | 1.35 (1.02 to 1.81) |
| RCS Charlson score† | | | | | | | | |
| 0 | | Ref | | Ref | | Ref | | Ref |
| 1 | | 1.17 (1.13 to 1.21) | | 1.13 (1.09 to 1.18) | | 0.89 (0.64 to 1.25) | | 1.27 (0.98 to 1.67) |
| 2+ | | 1.35 (1.26 to 1.44) | | 1.43 (1.32 to 1.55) | | 1.49 (0.79 to 3.19) | | 1.95 (1.31 to 3.03) |
| IMD quintiles | | | | | | | | |
| 1 | | Ref | | Ref | | Ref | | Ref |
| 2 | | 1.04 (1.01 to 1.08) | | 1.01 (0.97 to 1.05) | | 1.45 (0.98 to 2.15) | | 0.92 (0.70 to 1.20) |
| 3 | | 1.07 (1.04 to 1.11) | | 1.02 (0.97 to 1.06) | | 1.19 (0.80 to 1.78) | | 0.85 (0.64 to 1.12) |
| 4 | | 1.07 (1.03 to 1.11) | | 1.05 (1.00 to 1.10) | | 1.34 (0.90 to 2.02) | | 0.90 (0.65 to 1.25) |
| 5 | | 1.11 (1.06 to 1.16) | | 1.12 (1.06 to 1.18) | | 0.94 (0.53 to 1.79) | | 0.99 (0.70 to 1.45) |

Explanatory factors' relative effect (RE) on days of length of stay with 95% CI from univariable and multivariable regressions.
*Indicates knots for restricted cubic spline.
†Rheumatoid arthritis (RA) codes were omitted from the calculation of the Royal College of Surgeons (RCS) Charlson score and groups 2 and 3+ combined into 2+.
IMD, Index of Multiple Deprivation; THR, total hip replacement; TKR, total knee replacement.

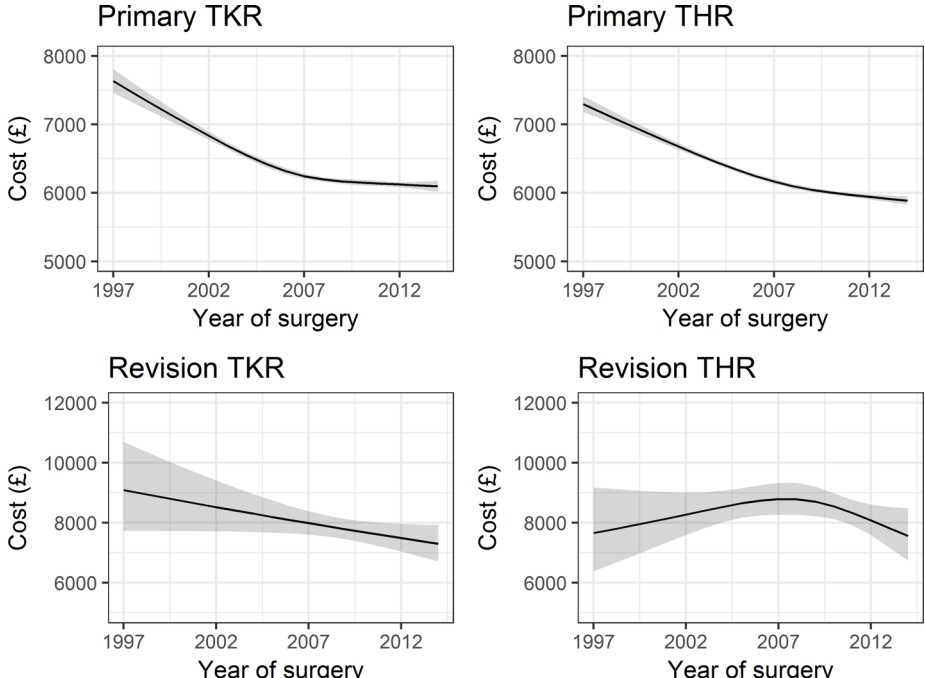

**Figure 3** Trends in hospital reimbursement. Estimated effect of year of surgery on hospital reimbursement (at 2016/2017 rates in GBP). THR, total hip replacement; TKR, total knee replacement.

## DISCUSSION
### Principal findings
Length of stay for primary TKR and primary THR fell significantly in the English NHS between 1997 and 2014. Expected length of stay fell by 11 days for primary TKR and 9 days for primary THR. The trajectory of the fall in expected length of stay was estimated as linear for primary THR, but had slowed somewhat towards the end of the study period for primary TKR. Length of stay also fell significantly for revision TKR, by 19 days between 1997 and 2014. Although a drop of 6 days was estimated for revision THR, this fall was not statistically significant.

Reductions is length of stay have, all else being equal, led to significant reductions in hospital reimbursement. Reimbursement was reduced by £1537 for primary TKR and £1412 for primary THR than otherwise would have been the case. The trajectories of these reductions in reimbursement were broadly in line with those of length of stay. The estimated reductions in reimbursement due to reduced length of stay for the revision procedures, £1792 for revision TKR and £91 for revision THR, were not statistically significant.

Age at surgery was associated with increased length of stay and reimbursement for each procedure with, in general, greater increases among older patients. RA and other comorbidities were associated with significantly higher length of stay and hospital reimbursement. Lower socioeconomic status was associated with a significant increase in length of stay and reimbursement for primary procedures. Male gender was associated with significantly shorter stays for primary procedures.

While the age, gender and socioeconomic status of those receiving joint replacement and revision surgery remained stable over time, individuals undergoing surgery had more comorbidities in more recent years, although the proportion diagnosed with RA fell. Controlling for patient characteristics had little effect on the association between year of surgery and length of stay or associated hospital reimbursement. These results imply that the downward trends in length of stay and associated hospital reimbursement are generally explained by increased efficiency.

### Study findings in context
Previous studies have also found length of stay to have fallen over recent decades following TKR and THR.[5–11] The downward trends in length of stay appear to be primarily due to increased efficiency rather than changes in patient characteristics.

A key driver of the efficiency gains in joint replacement has likely been a move towards 'fast-track' arthroplasty.[22] Enhanced recovery programmes (ERPs) have become increasingly prominent in orthopaedic surgery, aiming to standardise routine perioperative care, reduce length of hospital stay and promote rapid recovery.[23] These programmes have been found to reduce length of stay for a range of surgical procedures,[24] including knee and hip replacement.[25] The implementation of ERPs has been associated with either equivalent[23 26] or improved health outcomes,[25] and there is little evidence that ERPs lead to an increase in readmission rates.[24 27]

While ERPs have gained prominence, there remains scope for further implementation. As demonstrated in our study, both older age at surgery and comorbidities are associated with higher length of stay following TKR and THR. ERPs have been demonstrated to be appropriate for such patients, and the reductions in length of stay are, in fact,

**Table 3** Multivariable regressions for hospital reimbursement

| | Primary TKR (RE (95% CI)) | | Primary THR (RE (95% CI)) | | Revision TKR (RE (95% CI)) | | Revision THR (RE (95% CI)) | |
|---|---|---|---|---|---|---|---|---|
| | Unadjusted | Adjusted | Unadjusted | Adjusted | Unadjusted | Adjusted | Unadjusted | Adjusted |
| Year of surgery | 0.98 (0.97 to 0.98) | 0.98 (0.97 to 0.98) | 0.98 (0.98 to 0.98) | 0.98 (0.98 to 0.98) | 0.99 (0.97 to 1.00) | 0.99 (0.98 to 1.00) | 1.02 (0.99 to 1.04) | 1.02 (1.00 to 1.04) |
| Year of surgery* | 1.02 (1.01 to 1.03) | 1.02 (1.01 to 1.03) | 1.01 (1.01 to 1.01) | 1.01 (1.01 to 1.01) | | | 0.97 (0.95 to 1.00) | 0.97 (0.94 to 0.99) |
| Year of surgery* | 0.95 (0.91 to 0.99) | 0.94 (0.91 to 0.98) | | | | | | |
| Age | 1.00 (1.00 to 1.00) | 1.00 (1.00 to 1.00) | 1.00 (1.00 to 1.00) | 1.00 (1.00 to 1.00) | 1.01 (1.00 to 1.01) | 1.01 (1.00 to 1.01) | 1.01 (1.00 to 1.01) | 1.01 (1.00 to 1.01) |
| Age* | 1.00 (1.00 to 1.00) | 1.00 (1.00 to 1.00) | 1.00 (1.00 to 1.00) | 1.00 (1.00 to 1.00) | | | | |
| Age* | 1.02 (1.00 to 1.03) | | | | | | | |
| Gender: male | 1.01 (1.00 to 1.01) | 1.01 (1.00 to 1.01) | 0.99 (0.98 to 1.00) | 0.99 (0.98 to 1.00) | 0.99 (0.92 to 1.07) | 0.99 (0.92 to 1.07) | 0.93 (0.86 to 1.00) | 0.93 (0.86 to 1.00) |
| Diagnosis: RA | 1.04 (1.03 to 1.05) | 1.03 (1.01 to 1.04) | 1.03 (1.01 to 1.05) | 1.03 (1.01 to 1.04) | 1.37 (1.23 to 1.53) | 1.37 (1.23 to 1.53) | 1.13 (1.02 to 1.26) | 1.13 (1.02 to 1.26) |
| RCS Charlson score† | | | | | | | | |
| 0 | Ref | Ref | Ref | Ref | Ref | Ref | Ref | Ref |
| 1 | 1.05 (1.05 to 1.06) | 1.03 (1.02 to 1.04) | 1.03 (1.02 to 1.04) | 1.03 (1.02 to 1.04) | 1.04 (0.96 to 1.14) | 1.04 (0.96 to 1.14) | 1.10 (1.00 to 1.21) | 1.10 (1.00 to 1.21) |
| 2+ | 1.13 (1.11 to 1.15) | 1.10 (1.08 to 1.12) | 1.10 (1.08 to 1.12) | 1.10 (1.08 to 1.12) | 1.31 (1.10 to 1.58) | 1.31 (1.10 to 1.58) | 1.13 (0.97 to 1.32) | 1.13 (0.97 to 1.32) |
| IMD quintiles | | | | | | | | |
| 1 | Ref | Ref | Ref | Ref | Ref | Ref | Ref | Ref |
| 2 | 1.02 (1.01 to 1.03) | 1.02 (1.01 to 1.03) | 1.00 (0.99 to 1.01) | 1.00 (0.99 to 1.01) | 1.05 (0.95 to 1.16) | 1.05 (0.95 to 1.16) | 0.92 (0.84 to 1.02) | 0.92 (0.84 to 1.02) |
| 3 | 1.02 (1.01 to 1.03) | 1.00 (0.99 to 1.01) | 1.00 (0.99 to 1.01) | 1.00 (0.99 to 1.01) | 1.06 (0.96 to 1.18) | 1.06 (0.96 to 1.18) | 0.88 (0.80 to 0.98) | 0.88 (0.80 to 0.98) |
| 4 | 1.01 (1.00 to 1.02) | 1.01 (1.00 to 1.02) | 1.01 (1.00 to 1.02) | 1.01 (1.00 to 1.02) | 1.08 (0.97 to 1.20) | 1.08 (0.97 to 1.20) | 0.91 (0.81 to 1.03) | 0.91 (0.81 to 1.03) |
| 5 | 1.03 (1.02 to 1.04) | 1.02 (1.01 to 1.03) | 1.02 (1.01 to 1.03) | 1.02 (1.01 to 1.03) | 0.94 (0.81 to 1.10) | 0.94 (0.81 to 1.10) | 0.94 (0.82 to 1.08) | 0.94 (0.82 to 1.08) |

Explanatory factors' relative effect (RE) of diagnosis on hospital reimbursement (at 2016/2017 rates in GBP) with 95% CI from univariable and multivariable regressions.
*Indicates knots for restricted cubic spline.
† Rheumatoid arthritis (RA) codes were omitted from the calculation of the Royal College of Surgeons (RCS) Charlson score and groups 2 and 3+ combined into 2+.
IMD, Index of Multiple Deprivation; THR, total hip replacement; TKR, total knee replacement.

greater than for younger patients with fewer comorbidities.[28] Greater implementation of ERPs among such groups can, then, be expected to lead to further reductions in length of stay following joint replacement and additional cost-savings from the NHS perspective.

In the UK, government policy has also encouraged the use of specialist centres for joint replacement, and this has likely also contributed to the observed reductions in length of stay. Patients in specialised centres typically have lower length of stay.[9] This difference is not only due to patient selection as even after controlling for differences in the characteristics of those undergoing surgery in different centres, length of stay following THR is lower when surgery is undertaken at a specialist treatment centre.[15]

As would be expected, the reductions in length of stay translate into reduced reimbursement to hospitals than otherwise would be the case. This has also been seen in the USA where, although the overall costs for joint replacement have risen, falling length of stay has attenuated the increase.[29 30]

As observed here, the age and gender of those undergoing surgery have been relatively stable between 1991 and 2006.[3] Older age at surgery, particularly those over 80,[6 31] and of lower income[31] are both associated with longer stays. While gender has typically not been found to have an effect on length of stay[6] or costs,[32] being a woman was associated with longer stays following TKR in one study.[31] Over the study period, individuals appear to have an increasing number of comorbidities at the time of surgery for all of the procedures considered. This trend has previously been reported in the USA.[5 7 30] Comorbidities have been associated with longer hospital stays and greater costs following primary joint replacement.[31 32] The proportion of those undergoing primary TKR with a diagnosis RA though fell over time in this study. This trend has also previously been reported and is likely explained by therapeutic advances, such as the introduction of biologic therapies.[33] The relative impact of RA on length of stay and costs is not well known, but the increases estimated here may be explained by greater morbidity before surgery and the use of medications such as steroids and immunosuppressants.

### Strength and limitations of this study
Routinely collected data are not collected primarily to inform research, leading to a number of possible limitations. Coding accuracy is of particular concern. While the diagnosis of RA[34] and the coding of primary TKR and primary THR[35] have previously been validated in CPRD and HES APC, respectively, OA coding has not been thoroughly evaluated. If those misclassified as not having an OA diagnosis differed systematically from those with a recorded OA diagnosis, the external validity of our results may be affected.

A number of patient characteristics were controlled for to assess whether trends in length of stay and reimbursement were explained by changes in patient characteristics or improved efficiency. These included age, gender, comorbidities and socioeconomic status. However, other patient characteristics may have changed over time and influenced length of stay or reimbursement. Moreover, the changes in reported comorbidities may reflect changes in data capture over time.

A further limitation of this study is that we only examined the length of stay and hospital reimbursement of the spell for each individual's first recorded primary and revision procedure. This may have led to an underestimation of the total reimbursement to hospitals. In particular, revision procedures can have multiple stages and subsequent revision procedures may be required, which may lead to a higher overall hospital reimbursement than reported here. Furthermore, readmissions have not been included in the study. If reduced length of stay has been accompanied by an increase in the number of readmissions, then the reductions in resource use will have been overstated.

The primary strength of this study is that, despite potential concerns around coding accuracy, it was informed by routinely collected data, which provided real-world evidence of trends in length of stay. Sample sizes were large, particularly for primary TKR and primary THR, allowing a thorough analysis of trends over time, and the data were representative of the experience of the NHS in England.

The costs estimated here reflect reimbursement to hospitals by the NHS. By using current reimbursement rates, it has been possible to consider how changes in length of stay have translated into changes in reimbursement, all else being equal. However, trends in overall reimbursement likely follow a different trend given changes in other factors, such as the prices of implants. In addition, the accounting costs incurred by hospitals likely differ to the reimbursement received and have not been assessed in this study. Trends in the costs incurred by hospitals may differ, for example, if they are less sensitive to changes in length of stay.

### Policy implications and areas for future research
Expected length of stay associated with TKR and THR fell significantly between 1997 and 2014. This appears to be due to improved efficiency rather than changes in patient profiles. The reductions in length of stay have led, all else being equal, to reduced hospital reimbursement. If health outcomes have not worsened, reductions in length of stay will have improved the cost-effectiveness of joint replacement relative to what otherwise would have been the case.

With stretched healthcare budgets, further improvements in efficiency will likely be necessary for supply to keep pace with the rising demand for joint replacement procedures. Additional research into the drivers of the efficiency gains identified in this study would help inform future policy decisions.

**Acknowledgements** The authors would like to thank Miss Susan Thwaite (National Rheumatoid Arthritis Society) for her role as the patient and public representative and her role in the study steering committee. We also thank Dr Jennifer A de Beyer of the Centre for Statistics in Medicine, University of Oxford, for English language editing.

**Contributors** EB, CJE, DWM, AS, CC, NKA, RPV and DPA all made substantial contributions to conception and design of the study. EB, RPV and DPA undertook the statistical analysis. EB, RPV and DPA drafted the manuscript with CJE, DWM, AS, CC

and NKA revising it for important intellectual content. All authors read and approved the final manuscript.

**Funding**  DPA is funded by a National Institute for Health Research Clinician Scientist award (CS-2013-371 13-012). This work was supported by the NIHR Biomedical Research Centre, Oxford.

**Disclaimer**  This article presents independent research funded by the National Institute for Health Research (NIHR). The views expressed are those of the authors and not necessarily those of the NHS, the NIHR or the Department of Health.

**Competing interests**  All authors have completed the ICMJE uniform disclosure form at www.icmje.org/coi_disclosure.pdf and declare that NKA has received personal fees from Freshfields Bruckhaus Deringer, Bioventus, Flexion, Merck and Regeneron, all outside the submitted work. DPA reports grants from Amgen, Servier and UCB Biopharma, and non-financial support from Amgen, all outside the submitted work. DWM reports grants and personal fees from Zimmer Biomet. In addition, DWM has various patents related to Unicompartmental Knee Replacement (Zimmer Biomet) with royalties paid.

**Patient consent**  Not required.

**Provenance and peer review**  Not commissioned; externally peer reviewed.

**Data sharing statement**  CPRD data with HES linkage were provided under a licence that does not permit sharing. Data are obtainable from CPRD subject to a full application.

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
