## [Reviewer comments · BMJ Open]

ARTICLE DETAILS

TITLE (PROVISIONAL)	Trends and determinants of length of stay and hospital reimbursement following knee and hip replacement: evidence from linked primary care and NHS hospital records from 1997 to 2014
AUTHORS	Burn, Edward; Edwards, Christopher; Murray, David; Silman, Alan; Cooper, Cyrus; Arden, Nigel; Pinedo-Villanueva, Rafael; Prieto-Alhambra, Daniel

VERSION 1 – REVIEW

REVIEWER	William Brinson Weeks, MD, PhD, MBA The Dartmouth Institute Lebanon, NH 03766 USA
REVIEW RETURNED	28-Aug-2017

GENERAL COMMENTS	This is a well written and interesting paper that tracks patient characteristic, length of stay, and charge estimates for hip and knee replacement surgery (both primary and revision) for patients with osteoarthritis or rheumatoid arthritis in Britain in 1997-2014. The authors find that LOS decreased considerably, as did charge estimates, and that patient characteristics were largely similar over time, with the exception of numbers of comorbidities. The challenge that I find in the paper is the cost estimates. They are based on “2017/2018 draft prices” which includes a cost surcharge for “any stay longer than the trim-point, which is the upper limit of length of stay incorporated into the HRG cost.” So, there is some cyclical reasoning. The costs – which the authors note are “estimated costs [that] reflect current reimbursement to hospitals from the NHS, not the accounting costs incurred by hospitals or historical reimbursement rates” – then are driven by LOS. And, as the authors note (and has happened everywhere) LOS has dropped for these conditions considerably. My guess is that, if they existed at the time, ‘trim-points’ in 1997 were much, much higher than they were in 2014. And that fact likely drives the entire analysis, as the authors note that “high costs were driven primarily by stays longer than the trim-point.” Therefore, my recommendation is that the authors drop the cost analysis. It is pretty meaningless and might be better interpreted as reimbursement estimates applying the current reimbursement methodology to past time periods when different reimbursement methodologies were used.
---

	And it does not hold that decreased LOS means cost reductions. Because of high fixed costs in healthcare, a reduced LOS might save only a small amount (cost of disposable items) – unless staff were fired, hospitals were downsized, or wards were vacated. What LOS reduction does is increases capacity – which might be a good thing, but is not the same as reducing costs. If the authors are insistent on keeping the cost analysis, I would recommend that 1. They label them for what they are: NHS reimbursement estimates 2. They apply individual year reimbursement algorithms to each year, and then use a consumer price index to arrive at, say, 2014 pound equivalents, and 3. They contextualize the reimbursement estimates in some way (i.e., as a % of total NHS inpatient expenditures each year). The other revision that I'd suggest is an acknowledgement in the limitations that greater comorbidity may be a function of changes in data capture. Also, something that warrants explanation is why the numbers dropped so substantially in 2014 (from about 1000 TKA and THA between 2009-2013 that to a few hundred in 2014). I also think that the authors should elaborate on why ethical approval was "not required." Was the study found exempt from review?
--	--

REVIEWER	Christoffer Jørgensen Section for surgical pathophysiology Rigshospitalet, Copenhagen, Denmark I have received speakers fees from "Rapid Recovery" by ZimmerBiomet
REVIEW RETURNED	14-Sep-2017

GENERAL COMMENTS	The authors present the results of a large descriptive registry study on length of hospital stay and hospital costs in the U.K from 1997 to 2014. The study includes a large number of procedures, however considering that the timespan (21 years) of the study and 4 different procedures the individual number of procedures/year is not so impressive. Overall the research question is relevant for these procedures and the statistical method seems reasonable, however I am a little bit concerned about power, especially regarding RA patients and Revision procedures where several of the early years have 0-10 procedures/patients. The discussion is sound enough, but I am wondering about the complete lack of considerations regarding readmissions and the quite long length of hospital stay in elderly patients, even in the most recent years. Finally, one gets the impression that it is merely increased efficiency which may explain reductions in LOS. Rather there is a long list of studies suggesting that it is the combination of optimized logistics and more focus on using the most current evidence-based practice which reduces the surgical stress response and consequently the recovery period after these procedures (e.g. Husted et al. Acta Ort 2011 and Kehlet Lancet 2013). This reviewer certainly finds the study publishable, but it would benefit from a more nuanced discussion and also a focus on those patients who remain with a quite long LOS. Currently it mostly appears to be an epidemiological drill arguing improved efficiency, but with little discussion into the whys and how's.
--

Major comments

Article Summary

P3L17 Bullet one: What is “real-world evidence”? The use of the word evidence is misleading in this sentence. Rephrase into something like: Use of routinely collected data provide real-world information on... Furthermore, it is a bit ironic that what is summarized as a strength is later described initially as a limitation (P10L26) and then as a strength (P10L55). If what you mean is that these data may provide more accurate information on everyday clinical practice, as opposed to data from strictly controlled clinical trials then say so.

Introduction P4L29-38 I am puzzled by the complete lack of mention of some of the most recent studies (including studies published in BMJ-OPEN) on the influence of patient characteristics when using enhanced recovery protocols with short LOS (median 2-3 days), most of which have found that the impact of comorbidity may be less when adhering to such protocol.

Methods

As far as I can see you do not provide any data on the number of arthroplasty departments and how many procedures each accounted for? To my knowledge there is quite a bit of variance between NHS hospitals regarding LOS after these procedures, especially in older patients (see Starks et al. age and ageing 2014) could this not have influenced the results? As one of the primary outcomes is hospital costs, would it not be of interest to see whether this trend applied to all institutions? Such considerations should at least be included in the discussion.

Do you have any data on readmissions? These would be of interest as critics may claim that the reduction in initial hospital costs would be nullified by increased costs from readmission. Again you may want to include this as a potential limitation

P7L33 I have some difficulty with the use of Charlson score but omitting RA. I am aware that it is a recognized epidemiological way of adjusting for comorbidity, however is it not a bit problematic when omitting RA? Thus, a patient with RA +1 Charlson-score and myocardial infarction would not classify as Charlson 2+? However a patient with DM without organ failure and ulcer disease would, despite this may be of much less relevance than the RA. Could this not influence the estimates for Charlson 2+ on LOS and costs?

Results

I am puzzled by the very few revision TKR and THRs? In my country it is generally agreed that there is about 10% revision surgeries out of all procedures. How come you data includes so few? This cannot be explained by the inclusion of the first procedure only.

Appendix

Table on procedures.

How come the number of included procedures increases so drastically through the years, is it because there is an increase in reporting to the registry? Does it not skew the results to have a graduate increase from 78 to +700 procedures from 1997 to 2005 but about 1000 procedures from 2007-2013?

Discussion

P10L3 I really think this part of the discussion regarding comorbidities and age needs elaboration as it is the essence for further improvement. Suggesting you read and discuss the results of Starks et al Age and Ageing 2014 and Pitter et al Anaesth & Analg 2016 on the eldest patients. Both studies demonstrated that a LOS of 3-5 days is feasible in patients >85 years, however the Study by Pitter also documented areas for further improvement.

	Most importantly, both studies used enhanced recovery protocols, including both optimized logistic and best evidence based practice. Thus, improved treatment and recovery may be at least as important as optimizing efficiency of operating theaters etc. Your own data actually supports that patient comorbidities may have reduced impact on the LOS in recent years as it declined despite increased comorbidities. Therefore I completely disagree with P10L21-23 that the results imply that the downward trends are generally explained by increased efficiency. P11L9 Please move the Study findings in context section up before the strengths and limitations section. The way it stands now is disjointed, especially considering the before mentioned text. P11L21-25 Why do you mention unicompartmental knee arthroplasty? It is a different procedure and is not included in your data so it has no place here. Neither does minimally invasive surgery as it has never been well established that it provides benefits within these procedures. P11L24-30 The mention of enhanced recovery is very brief and should include the before mentioned considerations on the rationale behind such protocols (i.e. providing best evidence-based care and thus reducing the surgical stress response etc.) Furthermore, at least a few sentences on the effect of comorbidities when using such protocols should be mentioned when discussing this on P11L44-49. Limitations P10L46 Please include before mentioned considerations on readmissions and sample sizes by year/procedures/age. Minor comments P12L28 By the last sentence it seems as if the reduction in LOS is a goal by itself. In my opinion this is not so, rather it is a side-effect of improving patient care. Thus, interventions should not be aimed at reducing LOS, but at reducing postoperative morbidity which may increase LOS or readmissions. References: The reference list stops at ref 35, but 41 references are mentioned in the text? The references do not appear to be formatted after BMJ-OPENS style. Reference 5 and 7 are the same.
--	---

REVIEWER	Thomas Poder CIUSSS de l'Estrie - CHUS
REVIEW RETURNED	04-Oct-2017
GENERAL COMMENTS	I only reviewed the statistical methods as requested by the editor. This section is satisfactory and considers key points for the analysis of the study.

REVIEWER	Timo-Kolja Pfortner Institute for Medical Sociology, Health Service Research and Rehabilitation Science, Cologne, Germany
REVIEW RETURNED	05-Oct-2017

GENERAL COMMENTS	I was asked for doing a special statistical review of this manuscript, and have only one question regarding the inclusion of "explanatory variables". Don't you think that it is more appropriate to test for interactions (trends*explanatory variables) to test whether the main effect of year changes when including interaction terms? I have my doubts that you will find any changes in trend effects when including explanatory variables holdig at their statistical average.
--

VERSION 1 – AUTHOR RESPONSE

Reviewer: 1

Reviewer Name: William Brinson Weeks, MD, PhD, MBA Institution and Country: The Dartmouth Institute, Lebanon, NH 03766 USA Please state any competing interests or state 'None declared': None declared

Please leave your comments for the authors below see attached file

• We have addressed these comments below.

Reviewer: 2

Reviewer Name: Christoffer Jørgensen
Institution and Country: Section for surgical pathophysiology, Rigshospitalet, Copenhagen, Denmark
Please state any competing interests or state 'None declared': I have received speakers fees from "Rapid Recovery" by ZimmerBiomet

Please leave your comments for the authors below All comments are in the attached file

• We have addressed these comments below.

Reviewer: 3

Reviewer Name: Thomas Poder
Institution and Country: CIUSSS de l'Estrie - CHUS Please state any competing interests or state 'None declared': None

Please leave your comments for the authors below I only reviewed the statistical methods as requested by the editor.

This section is satisfactory and considers key points for the analysis of the study.

Reviewer: 4

Reviewer Name: Timo-Kolja Pfortner
Institution and Country: Institute for Medical Sociology, Health Service Research and Rehabilitation Science, Cologne, Germany Please state any competing interests or state 'None declared': None declared

Please leave your comments for the authors below I was asked for doing a special statistical review of this manuscript, and have only one question regarding the inclusion of "explanatory variables". Don't you think that it is more appropriate to test for interactions (trends*explanatory variables) to test whether the main effect of year changes when including interaction terms? I have my doubts that you will find any changes in trend effects when including explanatory variables holdig at their statistical average.

- We have estimated the univariable model with only year of surgery included to describe trends in length of stay and costs over time. We then have added patient characteristics to a multivariable model to estimate whether the effect of year of surgery was explained by changes in patient characteristics over time (if the estimated effect of year of surgery remained similar in the univariable and multivariable model the implication is that the trend over time is not explained by changes in the characteristics of those undergoing surgery). A similar approach to ours has been used previously for considering whether differences in length of stay between types of providers are due to differences in efficiency or patient selection.[1] In the multivariable model we did test for interactions between year of surgery and each patient characteristic, but found none to be significant. We have added this to the text (lines 189-191 and 248-249).

Reviewer 1

This is a well written and interesting paper that tracks patient characteristic, length of stay, and charge estimates for hip and knee replacement surgery (both primary and revision) for patients with osteoarthritis or rheumatoid arthritis in Britain in 1997-2014.

The authors find that LOS decreased considerably, as did charge estimates, and that patient characteristics were largely similar over time, with the exception of numbers of comorbidities. The challenge that I find in the paper is the cost estimates. They are based on “2017/2018 draft prices” which includes a cost surcharge for “any stay longer than the trim-point, which is the upper limit of length of stay incorporated into the HRG cost.”

So, there is some cyclical reasoning. The costs – which the authors note are “estimated costs [that] reflect current reimbursement to hospitals from the NHS, not the accounting costs incurred by hospitals or historical reimbursement rates” – then are driven by LOS. And, as the authors note (and has happened everywhere) LOS has dropped for these conditions considerably. My guess is that, if they existed at the time, ‘trim-points’ in 1997 were much, much higher than they were in 2014. And that fact likely drives the entire analysis, as the authors note that “high costs were driven primarily by stays longer than the trim-point.”

Therefore, my recommendation is that the authors drop the cost analysis. It is pretty meaningless and might be better interpreted as reimbursement estimates applying the current reimbursement methodology to past time periods when different reimbursement methodologies were used.

And it does not hold that decreased LOS means cost reductions. Because of high fixed costs in healthcare, a reduced LOS might save only a small amount (cost of disposable items) – unless staff were fired, hospitals were downsized, or wards were vacated. What LOS reduction does is increases capacity – which might be a good thing, but is not the same as reducing costs..

If the authors are insistent on keeping the cost analysis, I would recommend that 1. They label them for what they are: NHS reimbursement estimates 2. They apply individual year reimbursement algorithms to each year, and then use a consumer price index to arrive at, say, 2014 pound equivalents, and 3. They contextualize the reimbursement estimates in some way (i.e., as a % of total NHS inpatient expenditures each year).

- The purpose of including hospital reimbursement in the study was to describe how changes in length of stay have translated into savings from an NHS perspective, all else being equal. We therefore estimated reimbursement at current levels with the estimated ‘costs’ being the reference costs from the NHS perspective (equal to their reimbursement to hospitals). We used current reimbursement schedules as this allows us to compare current NHS costs to what they would be if practice was unchanged. So, for example, the drop in estimated NHS costs of £1,537 for primary TKR over the study period reflects the saving to the NHS from changes in practice over time (i.e. if length of stay and patient profiles had not changed over time, the costs would have remained the same).

- We have clarified the rationale of why we estimated costs in this way (lines 108, 119-121, 272-273) and that they reflect costs from the NHS perspective which are equal to the reimbursement provided to hospitals. Throughout the manuscript we have changed 'hospital costs' to 'hospital reimbursement' to better describe our estimates.
- We have also added that trends in costs from the hospital perspective may have had a different trend over time and added this as a limitation to our study (lines 384-385).
- We have reported costs savings as the absolute estimate on a per-patient level as we believe this is most intuitive and interpretable.

The other revision that I'd suggest is an acknowledgement in the limitations that greater comorbidity may be a function of changes in data capture.

- We have added this to the limitations of our study (lines 380-381).

Also, something that warrants explanation is why the numbers dropped so substantially in 2014 (from about 1000 TKA and THA between 2009-2013 that to a few hundred in 2014).

- We have clarified that the APC HES data extract used started in April 1997 and finished in March 2014 (lines 132-134).

I also think that the authors should elaborate on why ethical approval was "not required." Was the study found exempt from review?

- We have clarified that approval for the study was granted by the CPRD Independent Scientific Advisory Committee.

Reviewer 2

Reviewers assessment:

The authors present the results of a large descriptive registry study on length of hospital stay and hospital costs in the U.K from 1997 to 2014. The study includes a large number of procedures, however considering that the timespan (21 years) of the study and 4 different procedures the individual number of procedures/year is not so impressive. Overall the research question is relevant for these procedures and the statistical method seems reasonable, however I am a little bit concerned about power, especially regarding RA patients and Revision procedures where several of the early years have 0-10 procedures/patients.

- Our analysis is informed by data on individuals who are in a primary care database, which covers around 7% of the UK population, have a record of an incident diagnosis of osteoarthritis (OA) or rheumatoid arthritis (RA), and then had a surgery of interest recorded in their linked hospital data. The study size was determined by the number of individuals who satisfied the inclusion criteria for the study.
- While numbers are small for revision procedures and for those with RA, our sample size was sufficient to test for important differences in costs and length of stay. Indeed, we found RA to have a statistically significant impact on length of stay following both revision TKR and revision THR. And although we do have smaller numbers in earlier years, a significant trend is estimated over time (although confidence intervals are larger for estimates for earlier years, as can be seen in the figures).

The discussion is sound enough, but I am wondering about the complete lack of considerations regarding readmissions and the quite long length of hospital stay in elderly patients, even in the most recent years.

- We have added a consideration of readmissions with respect to enhanced recovery programmes (lines 332). It was beyond the scope of our study to assess length of stay and the costs of all readmissions and we have added this as a limitation to our study (lines 387-390).
- We have also added further consideration of the high length of stay for elderly patients as suggested (lines 335-341)

Finally, one gets the impression that it is merely increased efficiency which may explain reductions in LOS. Rather there is a long list of studies suggesting that it is the combination of optimized logistics and more focus on using the most current evidence-based practice which reduces the surgical stress response and consequently the recovery period after these procedures (e.g. Husted et al. Acta Orth 2011 and Kehlet Lancet 2013).

- We agree that the reductions in length of stay observed in our study can, at least in part, be explained by the improvements in patient care, such as from the increased usage of enhanced recovery pathways. In our study we consider such changes to be drivers of improved 'efficiency'. That is, rather than patient profiles changing over time, reductions in length of stay are due to changes in the way surgery is provided. We have clarified that the efficiency gains are likely due to improvements in the provision of care (lines 322-330).

This reviewer certainly finds the study publishable, but it would benefit from a more nuanced discussion and also a focus on those patients who remain with a quite long LOS. Currently it mostly appears to be an epidemiological drill arguing improved efficiency, but with little discussion into the whys and how's.

- We have added further details on the potential drivers of increased efficiency (lines 329-332, 342-347) and on the potential for greater implementation of enhanced recovery pathways for patients who are older and have more comorbidities at time of surgery (lines 335-341).

Major comments

Article Summary

P3L17 Bullet one: What is "real-world evidence"? The use of the word evidence is misleading in this sentence. Rephrase into something like: Use of routinely collected data provide real-world information on...

- We have edited this bullet point as suggested.

Furthermore, it is a bit ironic that what is summarized as a strength is later described initially as a limitation (P10L26) and then as a strength (P10L55). If what you mean is that these data may provide more accurate information on everyday clinical practice, as opposed to data from strictly controlled clinical trials then say so.

- We have clarified that the use of routinely collected data has both advantages, in particular data reflecting routine practice (lines 391-395), and disadvantages, with uncertainties around the reliability of coding (lines 370-375).

Introduction P4L29-38 I am puzzled by the complete lack of mention of some of the most recent studies (including studies published in BMJ-OPEN) on the influence of patient characteristics when using enhanced recovery protocols with short LOS (median 2-3 days), most of which have found that the impact of comorbidity may be less when adhering to such protocol.

- We have added that enhanced recovery pathways have been found to be particularly effective for patients who are older and have more comorbidities at time of surgery (lines 335-339).

Methods

As far as I can see you do not provide any data on the number of arthroplasty departments and how many procedures each accounted for? To my knowledge there is quite a bit of variance between NHS hospitals regarding LOS after these procedures, especially in older patients (see Starks et al. age and ageing 2014) could this not have influenced the results? As one of the primary outcomes is hospital costs, would it not be of interest to see whether this trend applied to all institutions? Such considerations should at least be included in the discussion.

- It was beyond the scope of our study to assess whether changes in hospital characteristics (e.g. surgery volume, size and type of arthroplasty departments) explain the changes in LOS.

However, an increase in the proportion of specialist centres in the UK is another driver of the observed reduced length of stay. We have added further consideration of this (lines 342-341).

Do you have any data on readmissions? These would be of interest as critics may claim that the reduction in initial hospital costs would be nullified by increased costs from readmission. Again you may want to include this as a potential limitation.

- It was also beyond the scope of our study to assess whether reductions in length of stay have led to increased readmissions. We have added further consideration of this in the discussion, e.g. when referring to enhanced recovery protocols (line 333), and have added this as a limitation to our study (lines 387-389).

P7L33 I have some difficulty with the use of Charlson score but omitting RA. I am aware that it is a recognized epidemiological way of adjusting for comorbidity, however is it not a bit problematic when omitting RA? Thus, a patient with RA +1 Charlson-score and myocardial infarction would not classify as Charlson 2+? However a patient with DM without organ failure and ulcer disease would, despite this may be of much less relevance than the RA. Could this not influence the estimates for Charlson 2+ on LOS and costs?

- We have excluded RA from the Charlson score as RA is included in our regressions as an explanatory variable. If we included RA in the Charlson score we would be double counting its effect. This does, however, affect the interpretation of the effect of the Charlson score as it becomes a measure of comorbidities aside from RA in our study, rather than all comorbidities. We have added further explanation of rationale and the implications of this approach to the text (lines 198-199).

Results

I am puzzled by the very few revision TKR and THRs? In my country it is generally agreed that there is about 10% revision surgeries out of all procedures. How come your data includes so few? This cannot be explained by the inclusion of the first procedure only.

- The study size was determined by the number of individuals who satisfied the inclusion criteria for the study. The low number of revisions is a result of the requirements for inclusion in the study. For a revision to be included, an individual needed both their incident diagnosis of OA or RA and the revision procedure recorded within follow-up. Given the typical time between diagnosis to primary and then from primary to revision, the number of revision was therefore limited.

Appendix

Table on procedures.

How come the number of included procedures increases so drastically through the years, is it because there is an increase in reporting to the registry? Does it not skew the results to have a graduate increase from 78 to +700 procedures from 1997 to 2005 but about 1000 procedures from 2007-2013?

- This is because individuals required an observed incident diagnosis of OA or RA in primary care prior to surgery, for the surgery to be included. While this means there is more uncertainty around estimates for early years of follow-up, we do not believe this will have led to bias.

Discussion

P10L3 I really think this part of the discussion regarding comorbidities and age needs elaboration as it is the essence for further improvement. Suggesting you read and discuss the results of Starks et al Age and Ageing 2014 and Pitter et al Anaesth & Analg 2016 on the eldest patients. Both studies demonstrated that a LOS of 3-5 days is feasible in patients >85 years, however the Study by Pitter also documented areas for further improvement. Most importantly, both studies used enhanced recovery protocols, including both optimized logistic and best evidence based practice. Thus, improved treatment and recovery may be at least as important as optimizing efficiency of operating theaters etc. Your own data actually supports that patient comorbidities may have reduced impact on the LOS in recent years as it declined despite increased comorbidities.

Therefore I completely disagree with P10L21-23 that the results imply that the downward trends are generally explained by increased efficiency.

- We agree that the reductions in length of stay observed in our study can, at least in part, be explained by the increased use of enhanced recovery protocols. In our study we consider such policies to be a driver of improved 'efficiency'. That is, rather than patient profiles changing over time, reductions in length of stay are due to changes in the way surgery is provided.
- We have clarified in the text that improvements in patient pathways are likely the key driver of improved efficiency (lines 322-332).
- In our study we did not find that the effect of comorbidities on length of stay has changed over time as the estimated interaction term between the Charlson score and year of surgery was not significant.

P11L9 Please move the Study findings in context section up before the strengths and limitations section. The way it stands now is disjointed, especially considering the before mentioned text.

- We have moved this section as suggested.

P11L21-25 Why do you mention unicompartmental knee arthroplasty? It is a different procedure and is not included in you data so it has no place here. Neither does minimally invasive surgery as it has never been well established that it provides benefits within these procedures.

- We have removed this and focused instead on improvements in patient pathways which we agree has likely been the key driver in improved efficiency.

P11L24-30 The mention of enhanced recovery is very brief and should include the before mentioned considerations on the rationale behind such protocols (i.e. providing best evidence-based care and thus reducing the surgical stress response etc.) Furthermore, at least a few sentences on the effect of comorbidities when using such protocols should be mentioned when discussing this on P11L44-49.

- As suggested, we have added further context around enhanced recovery protocols and comorbidities (lines 326-351).

Limitations

P10L46 Please include before mentioned considerations on readmissions and sample sizes by year/procedures/age.

- We have added readmissions as a limitation (lines 387-390), but we believe our study was sufficiently powered to address our research questions.

Minor comments

P12L28 By the last sentence it seems as if the reduction in LOS is a goal by itself. In my opinion this is not so, rather it is a side-effect of improving patient care. Thus, interventions should not be aimed at reducing LOS, but at reducing postoperative morbidity which may increase LOS or readmissions.

- We have removed this sentence.

References:

The reference list stops at ref 35, but 41 references are mentioned in the text?

The references do not appear to be formatted after BMJ-OPENS style.

Reference 5 and 7 are the same.

- We have corrected the references.

1 Siciliani L, Sivey P, Street A. DIFFERENCES IN LENGTH OF STAY FOR HIP REPLACEMENT BETWEEN PUBLIC HOSPITALS, SPECIALISED TREATMENT CENTRES AND PRIVATE PROVIDERS: SELECTION OR EFFICIENCY? *Health Econ* 2013;22:234–42. doi:10.1002/hec.1826

VERSION 2 – REVIEW

REVIEWER	Christoffer Jørgensen Rigshospitalet, Denmark personal fee Rapid Recovery, Zimmer
REVIEW RETURNED	13-Nov-2017
GENERAL COMMENTS	Satisfactory changes and argumentations. No further comments